# Dissociable laminar profiles of concurrent bottom-up and top-down modulation in the human visual cortex

**Samuel JD Lawrence[1], David G Norris[1,2], Floris P de Lange[1]\***

[1]Donders Institute for Brain, Cognition and Behaviour, Radboud University Nijmegen, Nijmegen, Netherlands; [2]Erwin L. Hahn Institute for Magnetic Resonance Imaging, University Duisburg-Essen, Essen, Germany

**Abstract** Recent developments in human neuroimaging make it possible to non-invasively measure neural activity from different cortical layers. This can potentially reveal not only which brain areas are engaged by a task, but also how. Specifically, bottom-up and top-down responses are associated with distinct laminar profiles. Here, we measured lamina-resolved fMRI responses during a visual task designed to induce concurrent bottom-up and top-down modulations via orthogonal manipulations of stimulus contrast and feature-based attention. BOLD responses were modulated by both stimulus contrast (bottom-up) and by engaging feature-based attention (top-down). Crucially, these effects operated at different cortical depths: Bottom-up modulations were strongest in the middle cortical layer and weaker in deep and superficial layers, while top-down modulations were strongest in the superficial layers. As such, we demonstrate that laminar activity profiles can discriminate between concurrent top-down and bottom-up processing, and are diagnostic of how a brain region is activated.

DOI: https://doi.org/10.7554/eLife.44422.001

*For correspondence:
floris.delange@donders.ru.nl

## Introduction

Using 'ultra-high field' MRI systems of 7T and above, it has become possible to non-invasively measure fMRI responses at lamina-resolved spatial resolutions in humans (*Dumoulin et al., 2018*; *Koopmans et al., 2011*; *Polimeni et al., 2010*). This has allowed researchers to ask new questions about the functional organization of the human brain, and examine communication between brain areas in more detail than previously possible (*Kuehn and Sereno, 2018*). One important promise of laminar fMRI is its potential ability to distinguish between bottom-up and top-down BOLD responses. While these are spatially amalgamated at standard imaging resolutions (*Lawrence et al., 2017*; *Self et al., 2017*), they are expected to be expressed at different cortical depths. Bottom-up connections between brain areas are known to target the granular layer 4, at middle cortical depths, while top-down connections target deeper and superficial layers but largely avoid layer 4 (*Anderson and Martin, 2009*; *Felleman and Van Essen, 1991*; *Rockland and Pandya, 1979*). It should therefore be possible to tease apart the bottom-up and top-down contributions to a stimulus-driven BOLD response by examining that response across cortical depth.

Previous laminar fMRI studies suggest that this is indeed the case. For example, stimulus-driven responses in visual cortex have been shown to be strongest at middle depths (*Koopmans et al., 2010*), while top-down signals embodying contextual inference, prediction, attention and working memory operate at deep and/or superficial, but not middle, cortical depths (*Klein et al., 2018*; *Kok et al., 2016*; *Lawrence et al., 2018*; *Muckli et al., 2015*; *Scheeringa et al., 2016*). Similar results for top-down influences have also been reported for auditory (*De Martino et al., 2015*) and motor cortex (*Huber et al., 2017*). Whilst these results are encouraging, these studies have typically

**eLife digest** Recent advances in brain imaging have made it possible to map brain activity in areas of tissue less than a millimeter in size. This resolution offers particular advantages for studying the brain's outer surface, the cortex. The cortex is traditionally divided into several layers, each containing different types and arrangements of neurons. New high-resolution machines can now visualize the activity in individual layers of cortex, and this can reveal whether the layers also have different roles.

In humans, a large area in the cortex is devoted to vision. Our visual cortex receives sensory information that arrives from the eyes via the optic nerve. This is known as bottom-up processing. But what we see depends on more than just incoming sensory information: it also relies on where we focus our attention, and on our expectations about how things should look. Many optical illusions, for example, work because the brain attempts to decipher an ambiguous visual signal based on previous experiences. This use of existing knowledge to interpret sensory input is called top-down processing.

Using high-resolution brain scanning, Lawrence et al. show that bottom-up and top-down processing occur in different layers of visual cortex. Healthy volunteers viewed a series of images while lying inside a brain scanner. Lawrence et al. changed the contrast of the images to alter the volunteers' bottom-up processing: this affected activity in the middle layer of visual cortex. To adjust their top-down processing, the volunteers were asked to attend to different features of the images on different trials: these changes in attention had more effect in the layers on either side of the middle layer. This suggests that bottom-up processing occurs in the middle layer of visual cortex, whereas top-down processing takes place in the layers above and below.

The findings by Lawrence et al. will help to better measure activity in cortical layers using modern brain imaging techniques. With further technological improvements, it may become possible to image each layer in the brain in more detail, in particular for other areas that support complex cognitive processes.

DOI: https://doi.org/10.7554/eLife.44422.002

measured top-down signals in the absence of a bottom-up response. The rationale for this choice is clear, as bottom-up drive could affect all cortical layers due to quick communication between layers (*Self et al., 2013*) and blurring from spatial hemodynamics (*Uğurbil et al., 2003*; *Uludağ and Blinder, 2018*; *Yacoub et al., 2005*) could obscure layer-specific top-down effects. However, being constrained to measuring top-down responses in isolation limits the potential power of laminar fMRI experiments for exploring brain function. If the overall BOLD response to a stimulus could be separated into its bottom-up, stimulus-driven component and a top-down, modulatory component, this would open the door for increasingly complex task design in laminar fMRI experiments.

Here we measured lamina-resolved fMRI responses from human participants as they viewed visual stimuli and were required to attend to a specific stimulus feature (orientation). Our stimulus paradigm was designed to elicit concurrent bottom-up and top-down modulations of the stimulus-driven response through orthogonal manipulations of stimulus contrast (bottom-up) and feature-based attention (top-down), both of which are known to influence early visual cortex responses (*Boynton et al., 1999*; *Himmelberg and Wade, 2019*; *Kamitani and Tong, 2005*; *Martinez-Trujillo and Treue, 2004*; *Saenz et al., 2002*; *Treue and Martínez Trujillo, 1999*).

We predicted that response modulations driven by attention would operate at different cortical depths to those driven by changes in stimulus contrast. Specifically, contrast modulations were expected to be largest at middle cortical depths, as increases in contrast should be associated with stronger bottom-up input to the granular layer (*Hubel and Wiesel, 1972*; *Rockland and Pandya, 1979*). Top-down influences are generally expected to be strongest in the deep and/or superficial cortical depths (agranular layers, *Lawrence et al., 2017*). However, previous research into laminar effects of attention have been varied. Studies by *van Kerkoerle et al. (2017)* and *Klein et al. (2018)* report largely agranular effects of attention in V1 while others report effects in all layers of V1 (*Denfield et al., 2018*; *Hembrook-Short et al., 2017*). Further studies have also reported attention effects in all layers of V4 (*Nandy et al., 2017*) and the superficial layers of primary auditory cortex

(*De Martino et al., 2015*). Moreover, previous laminar attention studies have employed spatial or object-based attention, but the laminar circuits involved in feature-based attention have, to our knowledge, not yet been studied. It is therefore not clear whether we should expect feature-based attention to modulate responses in all layers or only agranular layers. Critically, both eventualities yield the prediction that modulations from feature-based attention should be more strongly expressed in agranular layers compared to those from stimulus contrast, which should only be strong in the granular layer.

To preview, we found that fMRI responses in the early visual regions (V1-V3) were strongly modulated by changes in stimulus contrast and feature-based attention, and that these effects were indeed expressed at different cortical depths. As predicted, attentional modulations were more strongly expressed in agranular layers, particularly the superficial layers, while stimulus contrast modulations were largest in the granular layer.

## Results

We report laminar-resolved fMRI responses from the early visual cortex (V1-V3) of 24 healthy human subjects while they viewed a series of plaid stimuli comprising two orthogonal sets of bars (one oriented 45°, hereafter referred to as 'clockwise', the other oriented 135°, hereafter referred to as 'counter-clockwise'; see *Figure 1*). Plaids were presented in blocks of 8 stimuli, during which participants monitored changes in bar width of either the clockwise or counter-clockwise bars (*Kamitani and Tong, 2005*). Importantly, both sets of bars varied in width independently of each other, meaning attention had to be focused on the cued orientation to succeed at the task. Stimulus contrast was also manipulated: each block of stimuli comprised either high (80%) or low (30%) contrast plaid stimuli. As such, for each stimulus block participants attended to either clockwise or counter-clockwise bars within high or low contrast plaid stimuli, which provided our top-down and bottom-up task manipulations, respectively.

### Bottom-up and top-down modulations of the BOLD response

Subjects were able to focus their attention on one set of oriented bars within a plaid and accurately discriminate changes in bar width between stimuli. On average, subjects performed at 83.5% correct (SD = 2.3) for low and 84.5% correct (SD = 1.5) for high contrast stimuli. Task difficulty was controlled by separate staircases for high and low contrast stimuli to match task difficulty across contrast levels. Despite this, the numerically small difference in task performance was significant (t [22]=2.52, p=0.019). To assess the effects of attention and stimulus contrast on brain responses, we divided visually active voxels within V1, V2 and V3 into subpopulations with a strong preference for clockwise orientations over counter-clockwise or vice versa (*Albers et al., 2018*; see Materials and methods). It was expected that voxels would respond more strongly during blocks in which their preferred orientation was attended, and that all voxels would respond more strongly to higher contrast stimuli.

As expected, BOLD responses in early visual cortex were modulated by both subjects' attention towards a specific orientation and changes in stimulus contrast (see *Figure 2*). Responses to high contrast stimuli were significantly higher than low contrast stimuli across V1-V3 (F [23, 1]=35.57, p=$4.00e^{-6}$). The size of this effect varied across areas (F [30.2, 1.3]=46.53, p=$1.77e^{-8}$), being larger in V1 than V2 and V3. Voxel responses were also higher when their preferred orientation was attended, compared to when the orthogonal orientation was attended (F [23, 1]=25.67, p=$4.00e^{-5}$). This effect also varied across visual areas (F [46, 2]=4.91, p=0.012), being slightly smaller in V1 compared to V2 and V3. Overall, therefore, our paradigm was successful in inducing strong modulations of stimulus-driven BOLD responses using bottom-up (contrast) and top-down (feature-based attention) task manipulations.

### Dissociable laminar profiles of bottom-up and top-down response modulations

Next, we determined whether the effects of feature-based attention and stimulus contrast on BOLD responses varied across cortical depth, and whether they did so differently from each other. To this end we computed separate BOLD time courses specific to three equal volume gray matter depth bins defining deep, middle and superficial cortex (*Lawrence et al., 2018*; *van Mourik et al., 2018a*, see Materials and methods for more information). Depth-specific time courses were normalized to

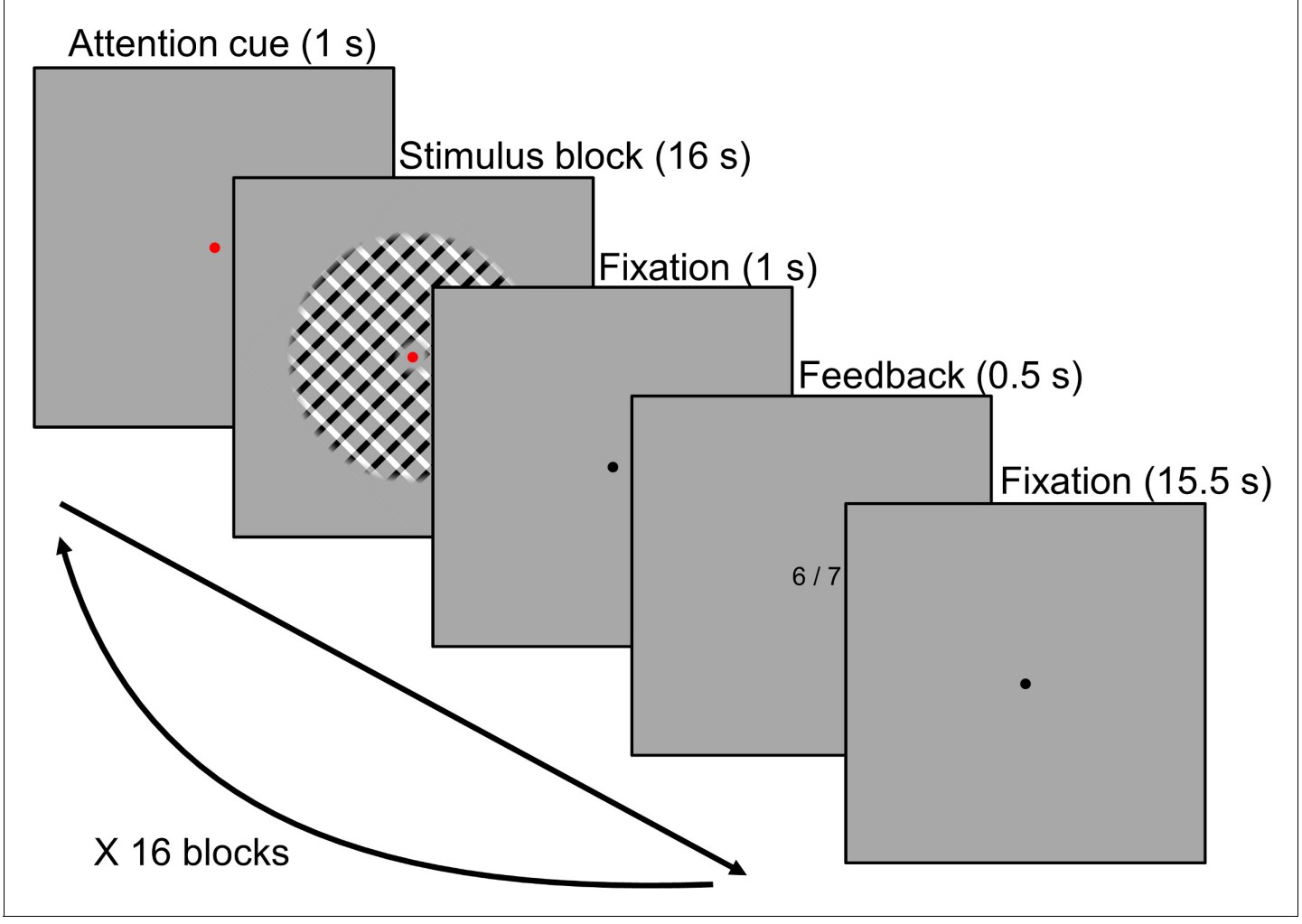

**Figure 1.** Task design. Plaid stimuli were presented in a block design. During stimulus blocks, eight stimuli were presented at a rate of 0.5 Hz (1.75 s on, 0.25 s off). Subjects were required to respond to each stimulus (except for the first in each block), indicating whether the bars in the cued orientation were thicker or thinner compared to the previously presented stimulus. Attention was cued by the colored fixation dot: red = clockwise, green = counter clockwise. Stimulus blocks were preceded by an attention cue and followed by performance feedback and an inter-block interval. See Materials and methods for more information on the task and stimuli.

DOI: https://doi.org/10.7554/eLife.44422.003

remove overall differences in signal intensity between layers (*Figure 3—figure supplements 3* and *4*). Note that this normalization was not critical to the results reported (*Figure 3—figure supplement 5*). Normalized depth-specific time courses were analyzed to compare the laminar profile of activity modulations resulting from top-down attention and bottom-up stimulus contrast. To get an overall picture of depth-specific modulations across the visual cortex, we first combined voxels from V1, V2 and V3 for this analysis.

Response modulations from both feature-based attention and stimulus contrast were clearly present in depth-specific time courses (*Figure 3A–D*). In order to fairly compare laminar profiles across conditions, we used data from the same time points (highlighted in *Figure 3A & C*), which comprised the peak of the BOLD response during a block of stimuli and during which both the effects of attention (F [23, 1]=19.95, p=1.76e$^{-4}$) and contrast (F [23, 1]=35.98, p=4.00e$^{-6}$) were significant. Within this time window, the effect of feature-based attention on neural responses was present at all cortical depths and was largest in the superficial layers (*Figure 3b*).

There was a trend of activity differences between layers induced by the attentional manipulation (F [46, 2]=2.82, p=0.070). Unpacking this, the attentional modulation was significantly stronger in the

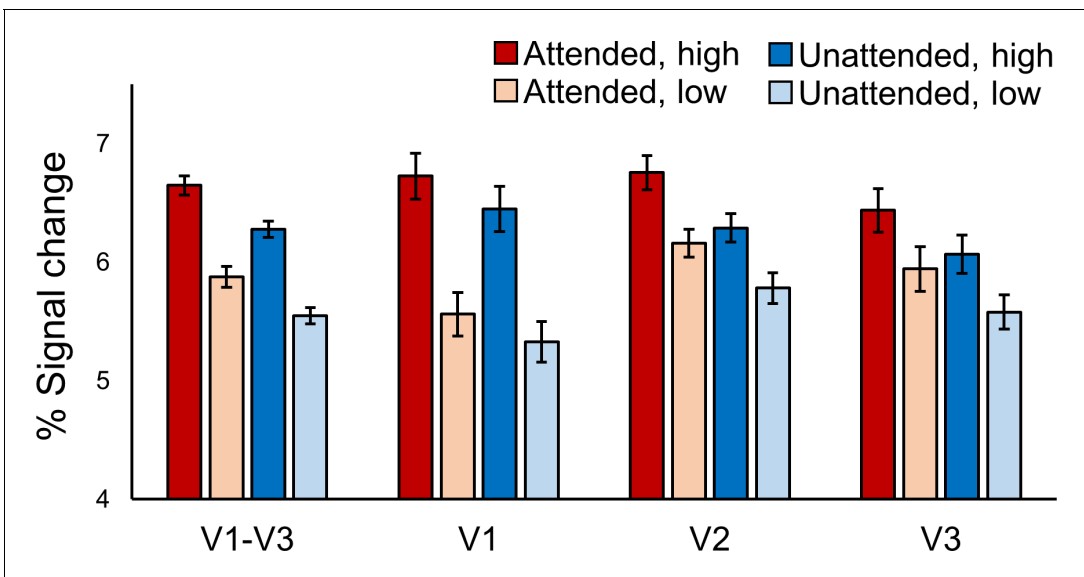

**Figure 2.** BOLD modulations from feature-based attention and stimulus contrast. Average BOLD signal change in orientation-selective voxels from V1-V3 combined, and V1, V2 and V3 separately. In all areas responses to high contrast stimuli (darker bars) were higher than to low contrast stimuli (lighter bars). Responses were also higher in voxels that preferred the orientation that was attended (red bars) compared to those that preferred the ignored orientation (blue bars). Error bars show within-subjects standard error. See text for statistical details.

DOI: https://doi.org/10.7554/eLife.44422.004

The following figure supplement is available for figure 2:

**Figure supplement 1.** Layer-specific BOLD signal changes for each experimental condition.

DOI: https://doi.org/10.7554/eLife.44422.005

superficial layers compared to the middle (t [23]=2.11, p=0.046) and deep layers (t [23]=2.15, p=0.042), while there was no significant difference in the strength of the attentional modulation between the deep and middle layers (t [23]=0.36, p=0.723). Modulations from changes in stimulus contrast were organized quite differently, peaking at middle depths (*Figure 3d*). Indeed, contrast modulations varied significantly across depth (F [46, 2]=8.43, p=0.001), being largest at middle compared to deep (t [23]=3.79, p=0.001) and superficial (t [23]=3.56, p=0.002) depths. Critically, the organization of contrast-related modulations across depth was significantly different to those caused by feature-based attention, as shown by a source (bottom-up, top-down) X layer (deep, middle, superficial) interaction (F [46, 2]=4.39, p=0.018). As such, the laminar profiles of responses modulations across the early visual cortex were dependent on whether those modulations were bottom-up or top-down in origin.

We predicted that top-down effects were more likely to be expressed in agranular layers compared to bottom-up effects. To explicitly test for this, we computed a score that described whether experimental effects were more agranular or granular. This was done by averaging the effect of feature-based attention (or contrast) from the superficial and deep depth bins (agranular) and subtracting that from the middle bin (granular). As such, a positive score indicates a largely agranular effect, while a negative score indicates a granular effect. As predicted, feature-based attention effects were more agranular compared to stimulus contrast (*Figure 3E*). This difference was significant (t [23] =3.11, p=0.005), and 20 of our 24 subjects showed an effect in this direction (*Figure 3F*). Therefore, it appears that top-down contributions to response modulations were stronger in the agranular layers compared to bottom-up contributions, which were strongest in the granular layer. As can be seen from *Figure 3B*, the agranular profile of attention was driven by the fact that the attentional modulation was strongest in the superficial layers.

## Laminar profiles are similar across early visual areas

We next explored how modulations from feature-based attention and stimulus contrast varied across cortical depth within visual areas, and potential differences in organization between areas. We

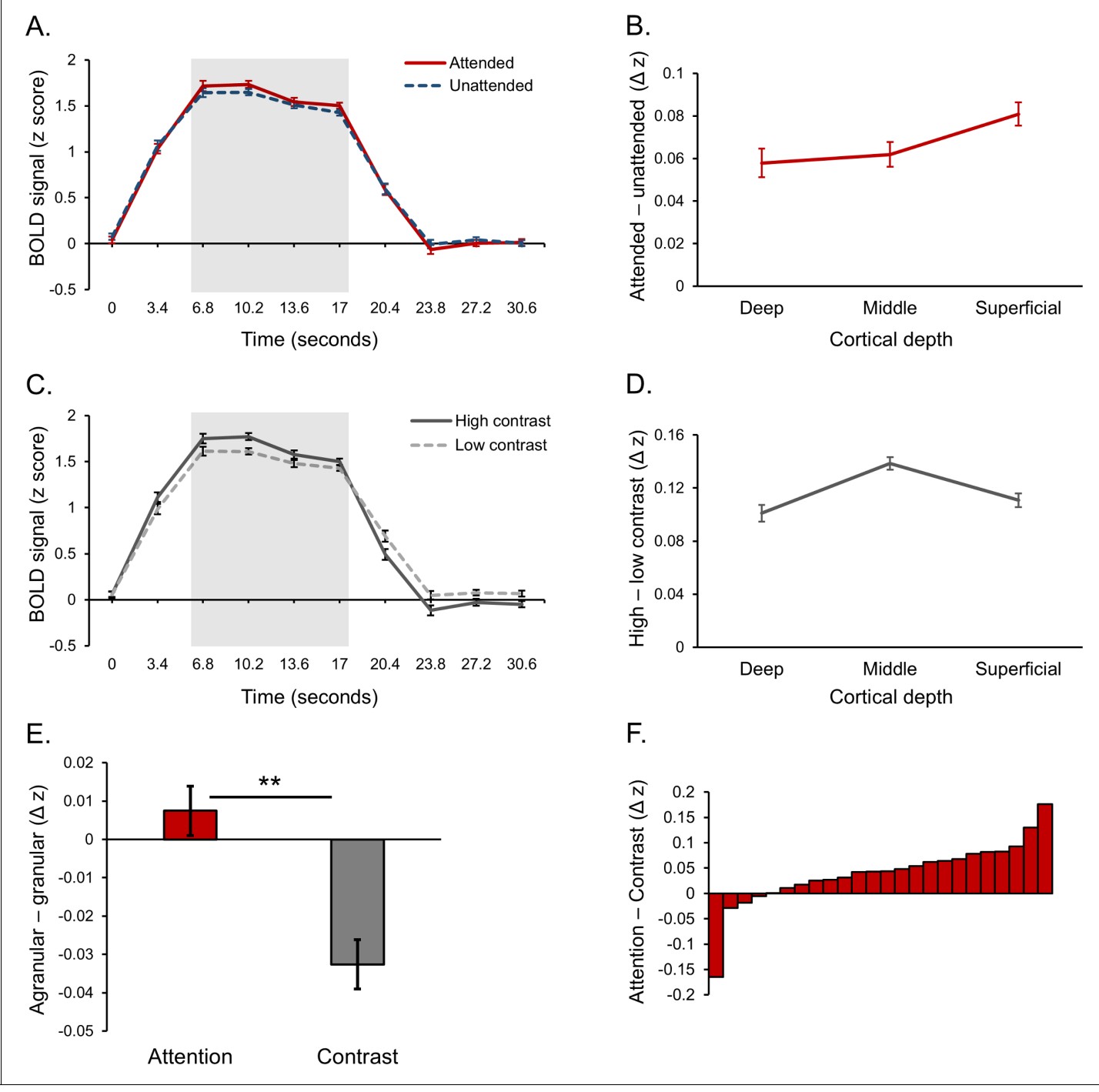

**Figure 3.** Laminar organization of top-down and bottom-up response modulations in V1-V3 combined. (**A**) Average BOLD time course for a block of stimuli, averaged across cortical depth bins. Responses from voxels that preferred the attended orientation (red line) and voxels that preferred the unattended orientation (blue dash) are plotted separately. (**B**) Average difference between attended and unattended BOLD signals from the highlighted time points in panel A, plotted separately for cortical depth bins. (**C**) Average BOLD time courses for blocks of stimuli contained high (dark gray line) and low contrast (light gray dash) stimuli. (**D**) Average difference between responses to high and low contrast stimuli from the time points highlighted in panel C, plotted separately for cortical depth bins. (**E**) Group average scores indicating the whether the effects of attention (red bar) and contrast (gray bar) were stronger in the agranular or granular layers. Scores were computed by taking the average attention or contrast effect from the deep and superficial layers (left-most and right-most data point in panels B and D, respectively) and subtracting the attention or contrast effect from the middle layer (middle data point in panels B and D, respectively). A positive score indicates a more agranular response, negative indicates more granular. Asterisks denote a significant paired sample t test (p=0.005, see text for details). (**F**) Difference between agranular – granular scores (panel E)

*Figure 3 continued on next page*

*Figure 3 continued*

for attention and contrast conditions for each individual subject. A Positive score indicates that attention modulations were stronger in agranular layers compared to contrast modulations, which was the case for 20 out of 24 subjects. All error bars show within-subject standard error.

DOI: https://doi.org/10.7554/eLife.44422.006

The following figure supplements are available for figure 3:

**Figure supplement 1.** Effect of mask size on results.

DOI: https://doi.org/10.7554/eLife.44422.007

**Figure supplement 2.** Results using re-sampled orientation preference masks designed to sample from all cortical depths equally.

DOI: https://doi.org/10.7554/eLife.44422.008

**Figure supplement 3.** Raw layer-specific time courses for each condition and visual area.

DOI: https://doi.org/10.7554/eLife.44422.009

**Figure supplement 4.** Removal of layer-specific BOLD bias.

DOI: https://doi.org/10.7554/eLife.44422.010

**Figure supplement 5.** Main results using raw layer-specific time courses with no normalization.

DOI: https://doi.org/10.7554/eLife.44422.011

**Figure supplement 6.** Differences in trial-to-trial variance between cortical layers.

DOI: https://doi.org/10.7554/eLife.44422.012

**Figure supplement 7.** Example cross section of V1 mask.

DOI: https://doi.org/10.7554/eLife.44422.013

**Figure supplement 8.** Layout of orientation-selective voxels in V1.

DOI: https://doi.org/10.7554/eLife.44422.014

**Figure supplement 9.** Example layer-specific masks and statistical maps.

DOI: https://doi.org/10.7554/eLife.44422.015

estimated depth-specific effects of attention and contrast for V1, V2 and V3 using the same methods applied to the three areas combined (Materials and methods). Similar to our original analysis, variation in the effect of attention across depth over V1-V3 (*Figure 4A*) did not reach significance (F [46, 2]=2.54, p=0.090), and attention depth profiles were similar across areas (F [69.67, 3.03]=0.44 p=0.778). The effect of contrast did vary across cortical depth (F [36.28, 1.58]=7.52, p=0.004) peaking at middle depths (*Figure 4B*), but this profile was not significantly different between the three areas (F [92, 4]=1.39, p=0.244. When directly contrasting these two modulatory factors, there was an overall, area independent, difference between feature-based attention and stimulus contrast that

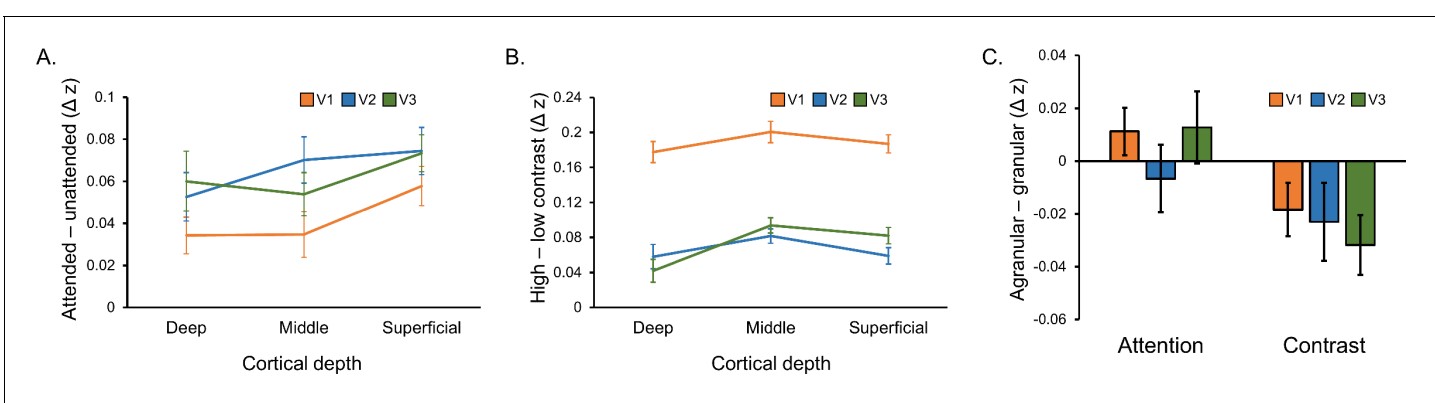

**Figure 4.** Laminar organization of top-down and bottom-up response modulations in V1, V2 and V3. (**A**) Average difference between depth-specific time courses in voxels that preferred the attended orientation and voxels that preferred the unattended orientation in V1 (orange), V2 (blue) and V3 (green). (**B**) Average difference between depth-specific time courses from blocks containing high and low contrast stimuli for V1, V2 and V3. (**C**) Average difference between attention modulations (taken from panel A) and contrast modulations (from panel B) in the agranular layers (average of deep and superficial bins) and granular layer (middle bin) for V1, V2 and V3. A positive score indicates a more agranular response, negative indicates more granular. All error bars show within-subject standard error.

DOI: https://doi.org/10.7554/eLife.44422.016

approached significance (F [46, 2]=2.74, p=0.075), but no significant differences between areas (F [80.38, 3.50]=1.00, p=0.407).

We also computed scores describing how agranular or granular effects of attention and contrast were within V1, V2 and V3 (*Figure 4C*). In general, modulations from feature-based attention were more agranular compared to those from stimulus contrast (F [23, 1]=5.48, p=0.028), and this was consistent across visual areas (F [46, 2]=0.51, p=0.607). Overall, these results show highly similar behavior of the three early visual regions (V1, V2, V3) that we examined, in terms of both their bottom-up and top-down laminar activation profiles.

## Discussion

We measured laminar fMRI responses from the human visual cortex during a visual task designed to induce bottom-up and top-down response modulations via orthogonal manipulations of stimulus contrast and feature-based attention. BOLD responses were strongly modulated by both feature-based attention and stimulus contrast, and these effects were expressed at different cortical depths. Effects of stimulus contrast were considerably larger at middle cortical depths compared to deep and superficial depths, while effects of feature-based attention were more even across depth, peaking in superficial cortex. Moreover, by comparing the strength of attention and contrast modulation in agranular versus the granular layers, we found that attention effects were expressed more strongly in the agranular layers (specifically the superficial layers) compared to effects from stimulus contrast, which were more granular.

Our results show clear differences in how bottom-up and top-down aspects of perceptual processing affect brain responses across cortical depth and are consistent with the anatomical organization of feedforward and feedback connections between brain areas (*Rockland and Pandya, 1979*). To our knowledge, our study also provides the first report of how visual cortex responses are modulated by feature-based attention at the laminar level. Most importantly, we demonstrate that laminar fMRI methods can be used to examine both the bottom-up and top-down components of the overall BOLD response as they co-occur during the processing of a stimulus. Previous laminar fMRI studies have either measured depth-specific effects in the absence of a physical stimulus (*Kok et al., 2016*; *Lawrence et al., 2018*; *Muckli et al., 2015*), or in the presence of a stimulus that was held constant (*De Martino et al., 2015*; *Klein et al., 2018*). By orthogonally manipulating stimulus contrast and feature-based attention, we have shown that top-down effects can be separated from concurrent bottom-up modulations driven by the stimulus. This opens the door for future studies to further examine the dynamic interactions between bottom-up and top-down processing that occur in the context of stimulus processing.

Top-down modulations of the BOLD response were expressed at all cortical depths relatively evenly, slightly peaking in the superficial layers. This partly contrasts with our previous study, which observed top-down activation of V1 during visual working memory that was strong in the agranular layers, but much weaker in the middle layer (*Lawrence et al., 2018*). The most obvious difference between the two studies is the presence of a physical stimulus during top-down modulation in this study, while there was no stimulus during working memory in our previous study. Each stimulus is expected to trigger a large response in the middle layer of V1 driven by bottom-up connections from the LGN (*Hubel and Wiesel, 1972*). Interestingly, influences of feature-based attention have been reported in the LGN before (*Ling et al., 2015*; *Schneider, 2011*), suggesting that this bottom-up signal could carry attentional modulations, consistent with our data. Electrophysiological studies of laminar effects of attention report mixed results regarding the involvement of the granular layer of V1 in attention. *van Kerkoerle et al. (2017)* report increased spike rate and current sinks with attention that were largest in the agranular layers. In particular, Van Kerkoerle et al. report strong attentional modulations in the deep layers compared to the middle layer, which was not the case in our data. Other studies (*Denfield et al., 2018*; *Hembrook-Short et al., 2017*) report attentional effects on spike rate in all layers of V1. However, it should be noted that these studies utilized spatial attention as opposed to feature-based attention, making it unclear how comparable their results are to our study. More research is required to further elucidate the laminar circuits involved in different modes of attentional control.

The relatively similar strength of attentional modulations across cortical layers highlights an important aspect of our task design. Taken in isolation, the laminar profile of feature-based attention

we report could be viewed as difficult to interpret, as there are no obvious differences between cortical depths. Crucially, however, the comparison of this profile to one derived from a manipulation of stimulus contrast revealed clear differences in how the visual cortex is modulated depending on the source of the modulation. We encourage future laminar studies exploring top-down responses to also include a bottom-up manipulation as a point of comparison, as the laminar organization of a BOLD activity difference between conditions on its own can be challenging to interpret (*Self et al., 2017*).

It is possible that we found top-down effects were similar across cortical depth due the blurring of BOLD responses across depth bins from spatial hemodynamics. Our task involved repeated presentation of a series of visual stimuli, which is expected to cause large swathes of stimulus-related activity in V1 that starts in layer four and quickly spreads to other layers (*Self et al., 2013*). This activity is in turn expected to be spatially blurred in the BOLD response by venous draining towards the pial surface, which smooths responses across cortical depth, causing stronger responses at superficial depths (*Uğurbil et al., 2003*; *Uludağ and Blinder, 2018*; *Yacoub et al., 2005*). It is therefore possible that repeated visual stimulation could have effectively washed out depth-specific responses, increasing the likelihood of experimental effects being uniform across depth. That said, any influence of spatial hemodynamics should be consistent across experimental conditions, and therefore accounted for in our calculation of bottom-up/top-down modulations via a subtraction of the responses to different contrast/attention conditions. Indeed, the strikingly distinct laminar profile of stimulus contrast effects that clearly peaked in the middle layers indicates that our analysis could account for the influence of hemodynamics. Nevertheless, accurate depth-estimates of BOLD responses continues to be the biggest challenge in laminar fMRI. Recent developments in modeling spatial aspects of the BOLD response for applying a spatial deconvolution to BOLD data (*Markuerkiaga et al., 2016*, ISMRM, abstract; *Marquardt et al., 2018*) and improved measurement protocols (*Huber et al., 2017*) could help to alleviate this issue.

We show that modulations of stimulus-driven responses were similar across areas within the early visual cortex. For stimulus contrast, this is consistent with a purely stimulus-driven effect that changes response amplitude at early, subcortical levels and is inherited through the visual system via bottom-up connections targeting layer 4 (*Hubel and Wiesel, 1972*; *Rockland and Pandya, 1979*). With regards to attention, there is little work addressing laminar differences between visual brain areas. *Nandy et al. (2017)* report attentional modulations in all layers of V4, consistent with our findings in extrastriate areas V2 and V3, as well as V1, but they do not provide a comparison to other brain areas. *Buffalo et al. (2011)* report attentional modulation of gamma and alpha oscillations in deep and superficial cortex that were similar in V1, V2 and V4. Though they did not measure from granular layer neurons, and thus cannot comment on whether attentional modulations occurred in all layers or only agranular layers, the similarity of results across visual brain areas appears consistent with our study. However, we again note that how these results compare to our own is unclear as these studies used spatial attention, not feature-based attention, as well as a variety of electrophysiological measurements with an unclear relation to the BOLD signal. For future studies, laminar fMRI is well suited to exploring laminar differences between brain areas as it affords simultaneous measurements over larger areas of cortex compared to electrophysiological methods.

In conclusion, we have shown that fMRI responses in visual cortex are strongly modulated by changes in stimulus contrast and feature-based attention, and that these effects operate at different cortical depths. Top-down modulations from attention were overall stronger in agranular layers (specifically the superficial layers) compared to those from stimulus contrast, which were strongest in the granular layer. We have shown that, in a task where bottom-up and top-down influences are manipulated independently, the overall BOLD response can be separated into top-down and bottom-up components by examining how these effects are organized across depth. Future studies can use similar strategies to further explore the dynamic interactions between bottom-up and top-down processing that occur in perception and cognition.

## Materials and methods

### Participants

Twenty-six healthy participants (all right-handed, nine males, mean age 25.5, age range 19–47) with normal or corrected-to-normal vision completed the experiment. This sample size (N = 26) provided us with 80% power to detect one-sided experimental effects that had at least medium effect size (Cohen's d > 0.6). All gave written informed consent and the study was approved by the local ethics committees (CMO region Arnhem-Nijmegen, The Netherlands, and ethics committee of the University Duisburg-Essen, Germany, protocol CMO 2014/288). Participants were reimbursed for their time at the rate of €10 per hour. All participants completed a 1 hr 3T fMRI retinotopic mapping session, a 1 hr psychophysics session, and a 1 hr 7T fMRI session for the main task. The experiment and analysis plan were preregistered on the Open Science Framework (https://osf.io/46adc/).

### Retinotopic mapping

Retinotopic mapping data were acquired and analyzed using identical methods to those reported in our previous study (*Lawrence et al., 2018*). In brief, brain responses to rotating wedge and expanding-ring checkerboard stimuli were acquired using a Siemens 3T Trio MRI system (Siemens, Erlangen, Germany) with a 32-channel head coil and a T2*-weighted gradient-echo EPI sequence (TR 1500 ms, TE 40 ms, 68 slices, 2 mm isotropic voxels, multi-band acceleration factor 4). One high resolution anatomical image was also acquired with a T1-weighted MP-RAGE sequence (TR 2300 ms, TE 3.03 ms, 1 mm isotropic voxels, GRAPPA acceleration factor 2). Anatomical data were automatically segmented into white matter, gray matter and CSF using FreeSurfer (http://surfer.nmr.mgh.harvard.edu/). Functional data were analyzed using the phase encoded approach in MrVista (http://white.stanford.edu/software/). Polar angle and eccentricity data were visualized on an inflated cortical surface and the boundaries of V1, V2 and V3 were drawn manually using established criteria (*Engel et al., 1994*; *Sereno et al., 1995*; *Wandell et al., 2007*).

### Psychophysics procedure

During the psychophysics session subjects completed the same visual task (*Figure 1*) that was used in the 7T main task fMRI session. Plaid stimuli were programmed in MATLAB (MathWorks, Natick, MA) and presented using PsychToolbox (*Brainard, 1997*) on a 24 inch BenQ XL2420T monitor (http://www.benq.eu/product/monitor/, resolution 1920 × 1080, refresh rate 120 Hz). Plaids comprised orthogonally oriented sets of bars (one set black, one set white), overlaid on top of each other. Areas of overlap between bars were made mid-gray (the same as the background), to facilitate mental separation of the two component stimuli. Subjects viewed the stimuli from a chin rest mounted 70 cm from the display and were instructed to fixate on a central fixation dot (0.5 degrees of visual angle across) at all times. Plaids were presented centrally behind an annulus mask (inner radius one degree, outer radius eight degrees, and had a spatial frequency of 1 cycle/degree and random phase. Stimulus edges were softened with a linear ramp that started 0.5 degrees from the edge of the mask.

The task used a block design. Stimulus blocks were preceded by an attention cue that lasted 2 s, where attention was cued by the color of the fixation dot (red = attend clockwise, green = attend counter-clockwise). The fixation dot remained red/green for the duration of the stimulus block. Stimulus blocks comprised a series of 8 plaid stimuli presented sequentially at a rate of 0.5 Hz (1.75 s on, 0.25 s off). Subjects' task was to press one of two buttons indicating whether the bars in the attended orientation were thicker or thinner than they were in the previously presented stimulus. Subjects were instructed to attend, but not respond to, the first stimulus in each block (as there was no preceding stimulus to compare to) and to respond to all remaining stimuli within the block. Subjects were allowed to respond at any time during stimulus presentation or the inter-stimulus interval, but the trial was marked as incorrect if they did not respond before the next stimulus was presented. Bar width for clockwise and counter-clockwise bars varied independently from each other, meaning attention had to be focused on the cued orientation in order to succeed at the task.

Changes in bar width between stimuli were controlled using a QUEST (*Watson and Pelli, 1983*) staircase function targeting 80% correct performance, which was updated after each individual trial. Bars within the first plaid presented in each block had a bar width of 0.2 degrees ± a random

increment between 0 and 0.02 degrees. For the remaining stimuli bar width was equal to the width of the previously presented stimulus ±an increment decided by the staircase. For both sets of bars, the direction of width increments was pseudo-randomized such that they were positive for four stimuli in each block and negative for four stimuli, presented in a random order. Stimulus luminance polarities were held constant within blocks but randomized between blocks, ensuring that both positive and negative luminance polarities were presented the same number of times for each experimental condition. After a stimulus block, the fixation dot turned black and was presented for 1 s. This was followed by performance feedback presented as a mark out of 7 for correct trials in the previous block, presented for 1 s. A 2 s attention cue then preceded the onset of the next stimulus block.

Subjects completed 24 blocks of the task, at which point the discrimination threshold for 80% correct performance was recorded for use in the main fMRI task. This process was performed once using high contrast stimuli (80% Michelson contrast), and once using low contrast stimuli (30% Michelson contrast), meaning separate thresholds were estimated for two contrast levels, which were used to match task difficulty across contrast levels in the fMRI experiment.

## fMRI data acquisition

fMRI data for the main experiment were acquired using a Siemens Magnetom 7T MRI system (Siemens, Erlangen, Germany) with a commercial RF head coil (Nova Medical, Inc, Wilmington, MA, USA) with one transmit (TX) and 32 receive (RX) channels and a gradient coil (Type AS095, Siemens Healthcare, Erlangen, Germany) with 38 mT/m gradient strength and 200 mT/m/ms slew rate. Functional data were acquired with a T2*-weighted 3D gradient-echo EPI sequence (*Poser et al., 2010*; TR 3408 ms, TE 28 ms, 0.8 mm isotropic voxels, 16˚ flip angle, 192 × 192×38.4 mm FOV, GRAPPA acceleration factor 4). Shimming was performed using the standard Siemens shimming procedure for 7T. Anatomical data were acquired with an MP2RAGE sequence (*Marques et al., 2010*; TR 5000 ms, TE 2.04 ms, voxel size 0.8 mm isotropic, 240 × 240 mm FOV, GRAPPA acceleration factor 2) yielding two inversion contrasts (TI 900 ms, 4˚ flip angle and TI 3200 ms, 6˚ flip angle), which were combined to produce a T1-weighted image. We also acquired a T2-weighted HASTE scan that was used to identify the calcarine sulcus to aid functional slice positioning (TR 3230 ms, TE 67 ms, seven coronal slices, 0.625 × 0.625×5.10 mm voxels). Stimuli were programmed and displayed using the same methods described for the psychophysics session onto a rear-projection screen using an EIKI (EIKI, Rancho Santa Margarita, CA) LC-X71 projector (1024 × 768 resolution, refresh rate 60 Hz), viewed via a mirror (view distance ~130 cm).

## fMRI procedure

Each subject completed 3 runs of the main task. The task was identical to the psychophysics session, except blocks of high and low contrast stimuli were randomly interleaved rather than presented in separate sessions, and timings were adjusted to sync with volume acquisition: The attention cue preceding a stimulus block was presented for 1.04 s, stimulus blocks lasted 16 s, followed by 1 s of fixation, 0.5 s of feedback, and a 15.54 s inter-block interval with a black fixation dot to allow the BOLD response to return to baseline before the next block. Changes in bar width for high and low contrast blocks were controlled by separate staircases, which were given starting estimates equal to contrast-specific discrimination thresholds measured in the psychophysics session plus a 20% increment. Due to a problem with recording button responses in one session, the behavioral results reported in the Results section were calculated using data from the remaining 23 subjects. five volumes were acquired per stimulus block, and five volumes between blocks, with 16 blocks in a single 555.5 s run. This run time also includes three dummy volumes that were discarded from the start of each run to allow for signal stabilization.

After the main task, subjects completed an orientation localizer scan that was used to measure voxel-wise orientation preference. Single sets of oriented bars (i.e., one stimulus component from the plaid stimuli presented in isolation) were presented in an AoBo block design. Stimulus blocks were 13.6 s long (4 TR) and separated by rest blocks of the same length. During a stimulus block bars were repeatedly presented with the same orientation at a rate of 2 Hz (250 ms on, 250 ms off). Stimuli were presented at 100% contrast, luminance polarity was reversed with each stimulus presentation, and phase was randomized. Stimulus blocks alternated between blocks of clockwise bars (45˚)

and blocks of counter-clockwise bars (135°). A total of 16 stimulus blocks were presented in a 446.4 s run; the first three volumes were again discarded. During the scan subjects maintained fixation and pressed a button every time the fixation dot flashed white for 0.25 s (1 to 4 flashes per block).

## Preprocessing of fMRI data

We used the same data processing pipeline as our previous study (*Lawrence et al., 2018*). Functional volumes were cropped so that only the occipital lobe remained, and spatially realigned within and then between runs using SPM8 (http://www.fil.ion.ucl.ac.uk/spm). Finally, data were highpass filtered using FEAT (fMRI Expert Analysis Tool) v6.00 (https://fsl.fmrib.ox.ac.uk/fsl) with a cut off of 55 s to remove low frequency scanner drift.

7T anatomical data were segmented into white matter, gray matter and CSF using FreeSurfer's (http://surfer.nmr.mgh.harvard.edu/) automated procedure. The white and gray matter surfaces were then aligned to the mean functional volume using a standard rigid body registration (*Greve and Fischl, 2009*) followed by a recursive non-linear distortion correction that has been described previously (*Lawrence et al., 2018*; *van Mourik et al., 2018a*).

## Definition of functional masks

We defined orientation-selective masks in V1, V2 and V3 using methods we have described previously (*Lawrence et al., 2018*). Note that the voxel selection procedure described here was applied to data from the orientation localizer scan; a data set that was independent from the main task. In brief, a GLM was applied to functional localizer data using FEAT v6.00 (https://fsl.fmrib.ox.ac.uk/fsl) to identify voxels that responded to significantly to all stimuli presented in the localizer (z > 2.3, p<0.05). For two subjects, there were very few voxels within the visual cortex that survived this cluster correction, indicating they had failed to remain alert for the duration of the experiment, and so we did not make any further use of their data. Next we contrasted responses to clockwise and counter-clockwise bars, and created masks of 1000 voxels per area, containing the 500 voxels with the most positive t values in this contrast (prefer clockwise) and the 500 with the most negative t values (prefer counter-clockwise). This was done separately for V1, V2 and V3. In any cases where there were fewer than 500 voxels within an area that met the required criteria for being visually active and having an orientation preference, we used as many voxels as did fulfill the criteria. To ensure that our results did not depend on how many and which voxels we chose to include on our masks, and that the selection we ran a battery of control analyses using an array of different mask sizes (*Figure 3—figure supplement 1–9*). Although effect sizes varied across mask sizes, all produced effects in the same direction as our main analysis.

## Quantification of effects of feature-based attention and stimulus contrast

Overall effects of feature-based attention and changes in stimulus contrast were quantified using a temporal GLM applied using FEAT v6.00 (https://fsl.fmrib.ox.ac.uk/fsl) on the preprocessed functional data. Each of the four experimental conditions (attend clockwise high contrast, attend clockwise low contrast, attend counter-clockwise high contrast, attend counter-clockwise low contrast) were modeled as separate regressors of interest and contrasted against baseline to estimate % signal changes associated with each condition. This was applied to orientation-selective masks from V1-V3 combined, and also to each area separately. % signal changes are shown in *Figure 2*. Signal changes associated with attended and unattended orientations were calculated by averaging responses from clockwise preferring voxels to attend clockwise blocks and counter-clockwise preferring voxels to attend counter-clockwise blocks. Likewise, unattended responses were calculated by averaging responses from clockwise preferring voxels to attend counter-clockwise blocks and vice versa.

## Estimation of laminar responses

Laminar-specific time courses were estimated using the open fMRI analysis toolbox (*van Mourik et al., 2018b*) as we have described previously (*Lawrence et al., 2018*). In brief, segmented cortical meshes were divided into five depth bins: white matter, three equivolume gray matter bins, and CSF. The proportion of overlap between each voxel within our orientation-selective masks and these

five bins were estimated, creating a matrix of depth weights describing the laminar organization of a population of voxels. These weights were regressed against the functional data from the same voxels to produce a single time course for each depth bin representative of the average response across the population at that cortical depth. This process was applied separately to the clockwise and counter-clockwise preferring voxel populations from V1-V3 combined to examine overall laminar activity across the visual cortex (*Figure 3*). We also did the same for V1, V2 and V3 separately to examine differences in laminar organization between areas (*Figure 4*).

## Normalization of layer-specific responses

It is well established that gradient-echo BOLD suffers from a bias in signal strength whereby responses in superficial cortex are stronger than responses from deep cortex (*Koopmans et al., 2010*; *Uğurbil et al., 2003*; *Uludağ and Blinder, 2018*; *Yacoub et al., 2005*). This bias can be seen clearly in our raw data (see *Figure 2—figure supplement 1*; *Figure 3—figure supplement 3*). We attempted to alleviate this issue by converting time courses specific to deep, middle and superficial cortical layers to z scores, normalizing differences in overall signal strength between layers. The z scoring was performed on layer-specific time courses from a single run of the main task, meaning it was performed within layers, across all experimental conditions and within runs. This procedure removed overall amplitude and variance differences between layers, while preserving within-layer differences between conditions. This had the effect of making overall signal changes between depth bins very similar (*Figure 3—figure supplement 4*), while preserving potential differences between depth bins that are due to experimental manipulations (rather than large differences in overall signal change). Of note, none of our results critically depend on this normalization step (*Figure 3—figure supplement 5*), but it allowed us to interpret those results in the absence of large-scale response differences between layers that are present in the raw data.

## Quantification of laminar-specific effects of feature-based attention and stimulus contrast

We analyzed time courses specific to deep, middle and superficial gray matter depth bins in the following way to quantify depth-specific effects of feature-based attention and stimulus contrast. Z scored, depth-specific time courses were split into segments of 10 volumes each, corresponding to one stimulus block (five volumes) followed by an inter-block interval (five volumes). To examine effects of feature-based attention, we computed an average attended time course by averaging responses for each block from the voxels that preferred the cued orientation in that block (i.e. prefer clockwise for attend clockwise blocks and prefer counter-clockwise for attend counter-clockwise blocks), and an average unattended time course by averaging responses from voxels that preferred the ignored orientation for each block (i.e. prefer clockwise for attend counter-clockwise blocks and prefer counter-clockwise for attend clockwise blocks). To examine effects of stimulus contrast, we averaged responses from both populations of voxels, regardless of orientation preference, averaging across all high contrast blocks and low contrast blocks to produce separate average time courses for high and low contrast stimuli. This analysis procedure was performed separately on time courses from the three gray matter depth bins within each subject, and then a group average was calculated. *Figure 3A&C* show group average time courses for each experimental condition, averaged across gray matter bins. The strength of modulations from feature-based attention and stimulus contrast were quantified as the difference between condition-specific time courses during the peak of the stimulus driven response (highlighted in *Figure 3A&C*), which are plotted for each depth bin in *Figure 3B&D*. Finally, we computed a score to describe the extent to which an effect of interest was expressed in the agranular or granular layers. This was achieved by averaging the effect of attention or stimulus contrast (*Figure 3B&D*) from the superficial and deep gray matter bins (agranular) and subtracting the middle bin (granular). A positive score therefore indicates a mostly agranular effect, while a negative score indicates a granular effect. The procedure described here was applied first to voxels from all visual areas combined (*Figure 3*), and then V1, V2 and V3 separately (*Figure 4*).

## Statistical testing

Overall effects of feature-based attention and stimulus contrast (*Figure 2*) were assessed using a visual area (V1/V2/V3) x contrast (high/low) x attention (attended/unattended) repeated measures

ANOVA. Note that, though we plot the results from V1-V3 combined in *Figure 2*, the ANOVA was performed on data from the three areas separately so that it would incorporate differences between areas.

The effects of feature-based attention and stimulus contrast in laminar-specific time courses from V1-V3 combined were quantified by examining the difference between attended and unattended (or high and low contrast) time courses during the peak of the stimulus-driven response during a block of stimuli (highlighted in *Figure 3A&C*). These were assessed using separate condition (attended/unattended or high/low contrast) x time point (6.8/10.2/13.6/17 s) repeated measures ANOVAs. These tests were performed on time courses averaged across depth bins that are plotted in *Figure 3A&C*. Depth-specific time courses were analyzed in the same way, and the difference between attended/unattended and high/low contrast for each depth are plotted in *Figure 3B&D*, respectively. We investigated whether these laminar profiles were different from each other using a modulation (attention/contrast) by depth (deep/middle/superficial) repeated measures ANOVA. A significant interaction (see Results) revealed the profiles were different from each other, so we examined them independently with one-way repeated measures ANOVAs (levels: deep/middle/superficial). In the cases that the main effect of depth was significant (i.e., for stimulus contrast), differences between depths were examined with paired-samples t tests. Finally, the difference between agranular – granular scores for attention and stimulus contrast was assessed using a paired-samples t test.

Laminar-specific effects of attention and stimulus contrast were also compared between visual areas (*Figure 4*). Differences between laminar profiles of attention and contrast and between areas were assessed with a visual area (V1/V2/V3) x modulation (attention/contrast) x depth (deep/middle/superficial) repeated measures ANOVA. The modulation x depth interaction approached significance (see Results), and we chose to examine the effects of attention and contrast using separate ANOVAs so that we might relate these results to those obtained from V1-V3 combined. As such, we examined whether the effects of attention and contrast varied across depth and visual area using separate visual area (V1/V2/V3) x depth (deep/middle/superficial) repeated measures ANOVAs. Finally, differences in agranular – granular scores between conditions and areas were assessed using a modulation (attention/contrast) x visual area (V1/V2/V3) repeated measures ANOVA. For all the ANOVAs we conducted, in cases where the assumption of sphericity was violated the degrees of freedom were adjusted using a Huynh-Feldt correction.

## Acknowledgements

We are grateful to Julia Pottkämper for valuable assistance in data collection and to Matthias Fritsche and Benedikt Ehinger for helpful comments on an earlier version of the manuscript. This work was supported by The Netherlands Organisation for Scientific Research (NWO Vidi grant 452-13-016) and the EC Horizon 2020 Program (ERC starting grant 678286, 'Contextvision'), both awarded to FPdL.

## Additional information

### Competing interests

Floris P de Lange: Reviewing editor, *eLife*. The other authors declare that no competing interests exist.

### Funding

| Funder | Grant reference number | Author |
| --- | --- | --- |
| Netherlands Organisation for Scientific Research | Vidi grant 452-13-016 | Floris P de Lange |
| European Research Council | Starting Grant 678286 CONTEXTVISION | Floris P de Lange |

The funders had no role in study design, data collection and interpretation, or the decision to submit the work for publication.

## Author contributions

Samuel JD Lawrence, Conceptualization, Resources, Data curation, Software, Formal analysis, Validation, Investigation, Visualization, Methodology, Writing—original draft, Project administration, Writing—review and editing; David G Norris, Resources, Supervision, Methodology, Writing—review and editing; Floris P de Lange, Conceptualization, Formal analysis, Supervision, Funding acquisition, Methodology, Writing—original draft, Project administration, Writing—review and editing

## Author ORCIDs

Floris P de Lange (iD) https://orcid.org/0000-0002-6730-1452

## Ethics

Human subjects: All participants gave written informed consent and the study was approved by the local ethics committees (CMO region Arnhem-Nijmegen, The Netherlands, and ethics committee of the University Duisburg-Essen, Germany). Protocol CMO 2014/288.

## Decision letter and Author response

Decision letter https://doi.org/10.7554/eLife.44422.021
Author response https://doi.org/10.7554/eLife.44422.022

# Additional files

## Supplementary files

• Transparent reporting form
DOI: https://doi.org/10.7554/eLife.44422.017

## Data availability

Data and code used for stimulus presentation and analysis are available online at the Donders Research Data Repository: https://data.donders.ru.nl/collections/di/dccn/DSC_3018028.04_752.

The following dataset was generated:

| Author(s) | Year | Dataset title | Dataset URL | Database and Identifier |
|---|---|---|---|---|
| Lawrence S, Norris DG, de Lange FP | 2019 | Dissociable laminar profiles of bottom-up and top-down modulation in the human visual cortex | https://data.donders.ru.nl/collections/di/dccn/DSC_3018028.04_752 | Donders Repository, DSC_3018028.04_752 |

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
