## [Decision Letter]

Thank you for submitting your article "Dissociable laminar profiles of concurrent bottom-up and top-down modulation in the human visual cortex" for consideration by *eLife*. Your article has been reviewed by three peer reviewers, including Christian Buchel as the Reviewing Editor and Reviewer #1, and the evaluation has been overseen by Michael Frank as the Senior Editor.

The reviewers have discussed the reviews with one another and the Reviewing Editor has drafted this decision to help you prepare a revised submission.

As you can see from the comments below the reviewers found the data and the study of high interest, However, they have also indicated that they need to see more details to fully judge the validity of the claims. This refers to the details of normalization (Reviewer #2) and overall quality control issues (Reviewer #3) including the wish to see more raw data (Reviewers #2 and #3). Reviewer #2 also questions whether the task employed is a classical feature-based attention task. This would also need to be addressed. Finally, Reviewer #3 wonders if the pre-selection of voxels is statistical valid ('double dipping').

Note from Deputy Editor: *eLife* prefers supplemental data to be supplements to specific figures, so that all data are included in the body of the paper. Please see if your existing supplemental figures can be thought of as supplements to figures in the paper, and whether the additional data asked for in the paper can also be included in that form. You will see that some *eLife* papers have 2 or 3 supplements to some figures, so that the relevant additional data are grouped with the primary result. And of course, there is nothing to prevent you from just adding additional figures to the paper. We prefer that controls be viewed as an important part of the paper.

*Reviewer #1:*

This is the first study combining high resolution, layer resolving fMRI at 7T with a task that allows to investigate top-down and bottom-up processing in a robust and elegant task. The methodology is sound and the authors have successfully established layer resolved fMRI. Based on retinotopically identified areas they investigate how attending to a particularly oriented grating CW or CCW is represented in early visual areas (V1, V2 and V3). The bottom-up manipulation was implemented by changes in contrast. The top-down manipulation was feature attention i.e. detecting changes in bar width. In a first step the authors identified voxels in the mapped areas that preferentially responded to either orientation. They then investigated both main effects i.e. high vs. low stimulus contrast and attended vs. unattended orientation. Finally, they analyzed whether these main-effects are stronger in a specific cortical layer (deep, middle and superficial). With respect to feature based attention they observed a trend that the superficial layer showed the strongest modulation, but this was not significant (p=0.07). In contrast, the bottom up contrast effect showed a significantly smaller main effect in the middle layer. They can also show a significant interaction (i.e. layer by type of attention). In a final step they investigated whether these effects differ between V1, V2 and V3. This analysis revealed no relevant differences in the observed patterns.

This is an interesting study showing that changes in stimulus contrast are predominantly represented in middle cortical layers. The study further suggests that feature based attention shows the opposite effect, yet this was only a trend. This a clear data set and very interesting result.

The paradigm employed is in essence a 2x2 factorial design with the factors bottom-up (i.e. stimulus contrast) and top-down (feature attention). Although Figure 2 suggests that there is no interaction, I was wondering whether (i) any voxels show such an interaction and (ii) whether this interaction would be differently expressed in different layers. Along these lines, Figure 4 collapses single main effects into a difference score, which does not allow the reader to interpret the full data. I agree that this might clutter the figure, but the authors should add a supplemental figure showing for each layer the responses to all 4 conditions without subtraction as a bar graph or in other words providing Figure 2 for each of the 3 layers.

*Reviewer #2:*

The authors present a 7T fMRI study examining whether top-down dependent processes (such as feature-based attention) can be dissociated from bottom-up processes (difference in contrast) in the different layers of early visual cortex. Many of the conclusions appear to rely on the computation of z-scores per layer and I have several methodological questions about the quantification, some of which may undermine the conclusions (but I hope the authors can address them – my point #1). I also have some issues with the behavioral paradigm (point #2) and whether this is truly a feature-based attention paradigm. The laminar differences in the attention effect are quite small but the comparison between agranular and granular layers might be possibly interesting. However, the authors use a distinction between granular and agranular (deep + superficial) in their analysis that does not describe the results well. The attentional effects are in fact strongest in the superficial layers and weakest in the deep layers, an intermediate in layer 4. It does therefore not make much sense to average across the superficial and deep layers, and the results are actually quite different from previous studies studying spiking activity in monkeys so that one wonders if laminar fMRI using the present methodology is a valid approach.

1) The authors indicate (subsection “Quantification of laminar-specific effects of feature-based attention and stimulus contrast”) that the BOLD signal in superficial layers is stronger than in the deeper layers, which is generally thought to be caused by the direction of blood flow in cortex. However, the raw data (before normalization) are not shown, and the normalization steps that carried out to correct for these differences in BOLD amplitude remained a bit obscure. This is an important point because I suspect that the laminar profile might actually reflect the choices that are made for the normalization.

As I understand it, the authors used the magnitude of the visually driven activity for normalization when they write "we converted time courses within depth bins to z scores". Does this imply that they normalized to the magnitude of visually driven activity per layer? If so, it seems somewhat surprising to see differences in visually driven activity between the layers, and in particular when considering Figure 3—figure supplement 4 suggesting that normalization was done per layer. I do understand that this can come out, because of the comparison between the activity elicited by high and low-contrast stimuli. But it is not immediately clear to me how one should interpret that difference, which should depend on the contrast response function of the voxels, not on visually driven response. Are the results in Figure 3—figure supplement 4 are obtained by pooling across the lower and higher contrasts? If the contrast response function is flat around the contrasts that are chosen, one might expect a small difference and a larger difference if the contrast response function is steeper. However, the authors seem to interpret the slightly stronger activity in layer 4 as evidence for a feedforward effect, and I am not sure if this interpretation is supported by the data, given these normalization issues.

– These normalization issues are aggravated when one also considers the s.d. (i.e. the variability) of activity across trials, a term appearing in the denominator when computing z-scores. Again, the outcome of the analysis may now become sensitive to arbitrary choices, which have not been well described. Are these z-scores computed per condition? Or across conditions? All conditions? In one possible scenario, the z-scores are computed across all stimuli (both low and high contrast stimuli- a similar argument holds for attended vs. non-attended stimuli). In that case, the outcome of the normalization depends on the effect size of the contrast manipulation which will contribute to the overall variance of activity across trials, and hence will contribute to the denominator when computing z-scores. In the most extreme case, contrast/attention explains a large fraction of the variance, and part of the effect of the contrast manipulation would be removed during the computation of z-scores because of the normalization. I am not sure if these problematic issues arose during the analysis, but the computation of z-scores and the effect sizes before normalization are not described in sufficient detail for a proper evaluation.

– The variance in activity might differ across the layers, have the authors also investigated these effects?

– If these normalization issues can be solved, which I hope to be the case, I would like to see a thorough discussion of a rational approach to normalize for the strength of BOLD signals in the different layers, how this affects the difference in activity elicited by low and high contrast stimuli, attended and non-attended stimuli and the possible issues that can occur when computing z-scores.

– I can imagine that systematic approaches to this problem must exist. If not, the authors may be in an excellent position to propose such an approach.

– I think it should be made clear already in the Results section that the laminar profiles have been z-scored within depth bins. This is important for interpretation of the results and on my initial reading I was wondering why there was no overall bias towards the superficial layers. I would also like to see a figure showing the non-z-scored BOLD signal changes for the different attention/contrast conditions to get a better impression of the data.

– In the Discussion the authors remark "That said, any influence of spatial hemodynamics should be consistent across experimental conditions, and therefore accounted for in our calculation of bottom-up/top-down modulations via a subtraction of the responses to different contrast/attention conditions." I was a bit confused, what do they mean with a subtraction?

– Is the increase in BOLD in the superficial layers a property of the chosen EPI sequence?

– Did the degree of orientation tuning differ between the areas. If yes, did that impact on the results?

– Figure 3F: not all subjects had more activity if the stimulus was attended. Is that a reliable within subject effect, opposite for some subjects than what was expected? Or does this reflect noise in the quality of the data of individual subjects?

2) The authors frame their paradigm as a feature-based attention paradigm. However, the design of the stimulus is quite different to a typical feature-based attention paradigm and I doubt whether the participants required feature-based attention to solve the task. For example, if I'm being cued to attend to the clockwise bars, my strategy would be to fixate one of the bars of the appropriate color (e.g. white) and monitor that bar for width changes. Eye-movements weren't monitored as far as I can tell, but even if the participants did maintain fixation then this is still more reminiscent of a spatial-attention task. True feature-based attention requires the participant to attend to a feature in one part of space, and this modulates activity related to that feature in another part of space. The interpretation of the paradigm is not just of semantic interest but has a large-impact on much of the discussion and the relevance and impact of this work, so I would like to hear the authors thoughts on this.

3) In the previous Current Biology paper, the orientation-mask had quite an interesting spatial distribution in V1. Are these the same subjects? If not, was the same distribution observed?

– In that paper in V1 bottom up signals are stronger in deep and superficial and weaker in L4 – their Figure 3B. Can you explain the difference?

4) I think that the pooling across deep and superficial layers and then report "agranular layers" is misleading, given that the results for layer 4 are more similar to the deep than to the superficial layers. The dissimilarity between the low attention modulation in the deep layers in the present study with the previous monkey studies should be discussed.

– The conclusion in the Discussion first paragraph "Moreover, our results pointed to stronger attention modulations in agranular cortical layers compared to contrast effects, which were strongest in the granular layer." seems to be a bit too optimistic to me.

*Reviewer #3:*

This study demonstrates clear modulations of relative layer-specific activation with a bottom up (contrast modulation) and top down (attention modulation) tasks. Specifically, in early visual regions, bottom up modulation was seen mostly in middle layers and top down modulation was mostly seen in superficial layers however with less modulation in middle layers. This is a well thought out experiment that certainly collapses challenging data in a manner that comes close to convincingly revealing the hypothezised modulations, however, as detailed in the specific comments, I have concerns with the fact that no actual maps nor raw BOLD activation laminar profiles nor even selected ROIs were shown, leaving the reader completely in the dark as to the data quality. It's mentioned that because there were two task modulations and a comparison between the two, large pial vessel effects were eliminated. This, I would argue is not entirely true as the baseline blood volume not only modulates the degree of BOLD signal change but can also have a secondary effect of enhancing the BOLD signal change difference as a function of underlying blood volume. Lastly, since the contrast modulation was used to create the ROI's and then used in the analysis, this paper has the potential for falling into the statistically unsound "double dipping" trap that would elevate the effects at least for the bottom up modulations. The way around this would be to demonstrate that these differences are mappable onto the cortical layer architecture. If this cannot be done, then it is questionable whether or not the data are of sufficient quality to make any conclusive statements.

[Editors' note: further revisions were requested prior to acceptance, as described below.]

Thank you for submitting your article "Dissociable laminar profiles of concurrent bottom-up and top-down modulation in the human visual cortex" for consideration by *eLife*. Your article has been reviewed by two peer reviewers, and the evaluation has been overseen by a Reviewing Editor and Michael Frank as the Senior Editor. The reviewers have opted to remain anonymous.

The reviewers have discussed the reviews with one another and the Reviewing Editor has drafted this decision to help you prepare a revised submission. As a consequence, I am delighted to say that we are happy, in principle, to publish a suitably revised version.

Essential revisions:

As you can see both reviewers still have issues with the paper. However, the editors feel that these concerns can be addressed by (i) a figure showing layer specific activation (Reviewer #3) (ii) by changing the wording as suggested by Reviewer #2 in critical parts of the manuscript, (iii) by further discussing the pooling of deep and superficial layers (Reviewer #2).

Should these revisions convince the editors, we are in principle happy to publish this paper,

*Reviewer #2:*

I am still a bit mixed about this paper. On the one hand, the authors convincingly addressed all issues that I had with the normalization.

On the other hand, my issue with the grouping of superficial and deep layers into an "agranular compartment" was not addressed satisfactorily. As I mentioned in my first review, the attentional effects (or the ratio with bottom-up effects) in fact seem strongest in the superficial layers and weakest in the deep layers, and intermediate in layer 4. It does therefore not make much sense to average across the superficial and deep layers. The results are actually quite different from previous studies on spiking activity in monkeys, so that one wonders if laminar fMRI using the present methodology reflects the underlying neurophysiology.

Several misleading statements remained in the revision:

– In the Abstract: "top-down modulation is significantly stronger in deep and superficial layers than top down effects", which is not shown for the deep layers but only by using the misleading grouping into "agranular layers". I suspect that the effects are driven by the superficial layers.

– Same at the end of the Introduction, final sentence.

– In reality there are no clear differences in top-down effects between the layers (subsection “Dissociable laminar profiles of bottom-up and top-down response modulations”) but it is only if a comparison (subtraction) is made to the bottom-up effects.

– This is also visible in Figure 4A, where the weakest attention effects are present in the deep layers, the strongest in the superficial layers and the granular layers are intermediate.

– Discussion section: "We have shown that, in a task where bottom-up and top-down influences are manipulated independently, the overall BOLD response can be separated into top-down and bottom-up components by examining how these effects are organized across depth." I think that this is an overstatement. The only reliable laminar difference seems to be in the bottom up response across layers.

– It should also be clarified consistently that these effects are driven by a difference in the contrast sensitivity rather than be a difference in the attention effects between layers.

What is the rationale of the grouping of superficial and deep layers? Is it the wish to replicate the non-human primate studies?

I would recommend that the three laminar compartments stay separate throughout the analysis (e.g. Figure 3E, F), and also in the Abstract and in the Discussion. It seems conceivable that in such an analysis with three laminar compartments, there is a difference in the ratio between top-down and bottom-up effects between superficial layers and the granular layers, but no such difference between the granular and deep layers. Such a discrepancy with the non-human primate work would also be a valuable outcome, and useful for future studies that plan to use laminar fMRI.

*Reviewer #3:*

Overall, I appreciate that the authors put in a tremendous amount of work to address all the questions from all three reviewers. I am satisfied with all the answers except this answer:

"Our revised manuscript includes three additional figures displaying raw data to allow the reader to better assess the data quality. (1) A figure showing raw, layer-specific BOLD time courses for each experimental condition and each ROI (Figure 3—figure supplement 3)"

This is a time course but not a map of actual layer activity.

"(2) A replication of our main results obtained by performing our analysis to raw data that had not been normalized, showing that the steps we took to minimize the impact of overall BOLD signal differences between layers, i.e. z scoring data within layers, were not critical to our results (Figure 3—figure supplement 5)."

This is certainly appreciated but not a map of layer specific activity.

"(3) A cross section of V1 from a representative example subject, where voxels are color coordinated based on which layer they belong to (Figure 3—figure supplement 7). We thank the reviewer for these helpful suggestions."

This is a mask and not an activation map of layer specific activity.

---

## [Author Response]

Reviewer #1:[…] This is an interesting study showing that changes in stimulus contrast are predominantly represented in middle cortical layers. The study further suggests that feature based attention shows the opposite effect, yet this was only a trend. This a clear data set and very interesting result.The paradigm employed is in essence a 2x2 factorial design with the factors bottom-up (i.e. stimulus contrast) and top-down (feature attention). Although Figure 2 suggests that there is no interaction, I was wondering whether (i) any voxels show such an interaction and (ii) whether this interaction would be differently expressed in different layers.

We thank the reviewer for this interesting suggestion. We ran an interaction analysis in FSL to probe whether there were voxels that showed an attention x contrast interaction. We found no evidence for significant clusters of voxels showing an interaction that replicated across participants, and no significant clusters at the group level. We therefore conclude that, as Figure 2 suggests, there were no voxels that exhibited an attention x contrast interaction.

Along these lines, Figure 4 collapses single main effects into a difference score, which does not allow the reader to interpret the full data. I agree that this might clutter the figure, but the authors should add a supplemental figure showing for each layer the responses to all 4 conditions without subtraction as a bar graph or in other words providing Figure 2 for each of the 3 layers.

We appreciate the suggestion and have added this figure to the supplementary materials (Figure 2—figure supplement 1).

Reviewer #2:[…] 1) The authors indicate (subsection “Quantification of laminar-specific effects of feature-based attention and stimulus contrast”) that the BOLD signal in superficial layers is stronger than in the deeper layers, which is generally thought to be caused by the direction of blood flow in cortex. However, the raw data (before normalization) are not shown, and the normalization steps that carried out to correct for these differences in BOLD amplitude remained a bit obscure. This is an important point because I suspect that the laminar profile might actually reflect the choices that are made for the normalization.As I understand it, the authors used the magnitude of the visually driven activity for normalization when they write "we converted time courses within depth bins to z scores". Does this imply that they normalized to the magnitude of visually driven activity per layer? If so, it seems somewhat surprising to see differences in visually driven activity between the layers, and in particular when considering Figure 3—figure supplement 4 suggesting that normalization was done per layer. I do understand that this can come out, because of the comparison between the activity elicited by high and low-contrast stimuli. But it is not immediately clear to me how one should interpret that difference, which should depend on the contrast response function of the voxels, not on visually driven response. Are the results in Figure 3—figure supplement 4 are obtained by pooling across the lower and higher contrasts? If the contrast response function is flat around the contrasts that are chosen, one might expect a small difference and a larger difference if the contrast response function is steeper. However, the authors seem to interpret the slightly stronger activity in layer 4 as evidence for a feedforward effect, and I am not sure if this interpretation is supported by the data, given these normalization issues.

We apologize for not adequately explaining the rationale and method for z scoring our data in the original manuscript. As has been documented before (Koopmans et al., 2010; Uǧurbil et al., 2003; Uludağ and Blinder, 2018; Yacoub et al., 2005), we found large differences in overall BOLD signal strength between cortical layers, where responses were strongest in superficial cortex. This can now clearly be seen in a new supplementary figure (Figure 3—figure supplement 3) which shows raw BOLD time courses for each experimental condition before any normalization. We wanted to investigate the strength of signal modulations from attention and stimulus contrast within layers but were not interested in differences in overall signal strength between layers. We therefore chose to normalize BOLD responses from each cortical layer separately. This was done by converting layer-specific time courses from each run of the fMRI experiment to z scores, meaning z scoring was performed within layers but across all experimental conditions. We reasoned that this should normalize differences in overall signal strength between layers, while preserving differences between conditions within layers. Z scoring within each run should also reduce the impact of differences in signal quality or intensity between runs. It is clear that this approach removed most of the overall signal intensity differences between layers by comparing the raw layer-specific time courses shown for each condition in Figure 3—figure supplement 3 to the normalized layer-specific time courses (averaged across conditions) in Figure 3—figure supplement 4. We now include new supplementary figure showing our main results without normalization, determined by repeating our analysis approach without converting layer-specific time courses to z scores (Figure 3—figure supplement 5). The final results look very similar to those reported in the main manuscript including the z scoring procedure. This shows that this normalization had little effect on layer-specific response differences between experimental conditions, but allows us to consider those differences in the absence of large differences in overall signal strength between layers, which are partly due to blood draining through the cortex (Uludağ and Blinder, 2018) rather than our experimental manipulations. We have now also added a new section to the Materials and methods that more clearly describes our reasonings behind the z scoring approach and how it was implemented (“Normalization of layer-specific responses”).

The reviewer also asks whether the time courses plotted in Figure 3—figure supplement 4 are averaged across contrast conditions, and whether voxel contrast response functions might be flat across the contrast levels we chose. The data in Figure 3—figure supplement 4 are indeed averaged across contrast conditions as well as across attention conditions. Although we did not measure full contrast response functions, we do not expect them to be flat across the contrast levels of 80% and 30% that we chose, given previously measured response functions (Buracas and Boynton, 2007). We found a strong effect of stimulus contrast on BOLD responses (Figure 2), and the effect of contrast can also be seen in the raw data by comparing time courses from low and high contrast conditions in Figure 3—figure supplement 3. We believe this provides sufficient evidence that voxel responses were modulated by stimulus contrast as we expected.

– *These normalization issues are aggravated when one also considers the s.d. (i.e. the variability) of activity across trials, a term appearing in the denominator when computing z-scores. Again, the outcome of the analysis may now become sensitive to arbitrary choices, which have not been well described. Are these z-scores computed per condition? Or across conditions? All conditions? In one possible scenario, the z-scores are computed across all stimuli (both low and high contrast stimuli- a similar argument holds for attended vs. non-attended stimuli). In that case, the outcome of the normalization depends on the effect size of the contrast manipulation which will contribute to the overall variance of activity across trials, and hence will contribute to the denominator when computing z-scores. In the most extreme case, contrast/attention explains a large fraction of the variance, and part of the effect of the contrast manipulation would be removed during the computation of z-scores because of the normalization. I am not sure if these problematic issues arose during the analysis, but the computation of z-scores and the effect sizes before normalization are not described in sufficient detail for a proper evaluation.*

We again apologize for not describing our z scoring procedure with sufficient clarity in the original manuscript. We hope that our manuscript revisions and response to point #1 have resolved this issue. We would again highlight that the replication of our main results without z scoring in Figure 3—figure supplement 5 should now make it clear to the reader that our choice of normalization was not critical to our results.

– The variance in activity might differ across the layers, have the authors also investigated these effects?

To interrogate this issue, we computed trial-to-trial standard deviation in our (normalized) layer-specific time courses for each of the 10 time points within a single trial, and then computed the average standard deviation across time points. The group average standard deviation for each layer (Figure 3—figure supplement 6) clearly shows a difference in overall signal variance between layers, where variance was higher in deeper cortex (F [46, 2] = 11.41, p = 9.5e-5). It therefore appears that signals we measured from deeper cortex had overall lower signal strength and larger variance. Given that our main results show an overall difference in the organization of bottom-up and top-down modulations across all layers, we do not believe higher signal variance in deeper cortex should greatly impact the interpretation of our data. In addition, the similarity of main results using raw (not normalized) data (Figure 3—figure supplement 5) to those using data that were z scored within layers (Figure 2) indicates that differences in variance between layers did not affect our calculation of average responses from the z scored data. Nevertheless, we now include this analysis in the supplementary as we consider it useful information for the reader and we thank the reviewer for the suggestion.

– If these normalization issues can be solved, which I hope to be the case, I would like to see a thorough discussion of a rational approach to normalize for the strength of BOLD signals in the different layers, how this affects the difference in activity elicited by low and high contrast stimuli, attended and non-attended stimuli and the possible issues that can occur when computing z-scores.– I can imagine that systematic approaches to this problem must exist. If not, the authors may be in an excellent position to propose such an approach.

We have included a new section to the Materials and methods, which describes our reasonings for the z scoring procedure and how it was implemented. This passage is also pasted below:

“Normalization of layer-specific responses

It is well established that gradient-echo BOLD suffers from a bias in signal strength whereby responses in superficial cortex are stronger than responses from deep cortex (Koopmans et al., 2010; Uǧurbil et al., 2003; Uludağ and Blinder, 2018; Yacoub et al., 2005). This bias can be seen clearly in our raw data (see raw time courses in Figure 3—figure supplement 3). […] Of note, none of our results critically depend on this normalization step (Figure 3—figure supplement 5), but it allowed us to interpret those results in the absence of large-scale response differences between layers that are present in the raw data.”

– I think it should be made clear already in the Results section that the laminar profiles have been z-scored within depth bins. This is important for interpretation of the results and on my initial reading I was wondering why there was no overall bias towards the superficial layers. I would also like to see a figure showing the non-z-scored BOLD signal changes for the different attention/contrast conditions to get a better impression of the data.

We now state more clearly in the Results section how the data have been normalized:

“Depth-specific time courses were normalized to remove overall differences in signal intensity between layers (see Figure 3—figure supplement 3 and 4). Note that this normalization was not critical to the results reported (Figure 3—figure supplement 5). Normalized depth-specific time courses were analyzed to compare the laminar profile of activity modulations resulting from top-down attention and bottom-up stimulus contrast.”

We also provide a new supplementary figure showing raw layer-specific time courses for each experimental condition and visual area (Figure 3—figure supplement 3).

– In the Discussion the authors remark "That said, any influence of spatial hemodynamics should be consistent across experimental conditions, and therefore accounted for in our calculation of bottom-up/top-down modulations via a subtraction of the responses to different contrast/attention conditions." I was a bit confused, what do they mean with a subtraction?

Here we refer to how we quantified the strength of modulations from stimulus contrast and feature-based attention. The effect of contrast was established by subtracting the averaged trial response to low contrast stimuli from the response to high contrast stimuli. Similarly, we quantified the effect of attention as the response to the unattended orientation subtracted from the response to the attended orientation. This procedure is described in the legend for Figure 2 and in the Materials and methods section (Quantification of effects of feature-based attention and stimulus contrast). We expect that the influence of spatial hemodynamics is consistent across contrast levels, and across attention conditions, meaning they should effectively be accounted for in this subtraction, leaving only the influence of contrast or attention.

– Is the increase in BOLD in the superficial layers a property of the chosen EPI sequence?

Indeed, increased BOLD strength in superficial cortex is a feature of gradient echo EPI. This has been reported in detail in previous research (e.g. Koopmans et al., 2010; Turner, 2002; Uǧurbil et al., 2003; Uludağ and Blinder, 2018; Yacoub et al., 2005). Other sequences such as spin echo EPI or 3D GRASE are less susceptible to this bias (De Martino et al., 2013; Uludaǧ et al., 2009), but they also provide much weaker overall signal-to-noise ratio (SNR) compared to gradient echo EPI (Moerel et al., 2017). Recent developments in laminar cerebral blood volume imaging appear very promising (Huber et al., 2017), but these are difficult to implement for the visual system and may also have lower SNR.

– Did the degree of orientation tuning differ between the areas. If yes, did that impact on the results?

Our experiment only included two orthogonal orientations, meaning we cannot extrapolate full orientation tuning profiles for each area. Instead we have examined the average selectivity of V1, V2, V3 for clockwise over counter-clockwise (or vice versa). This was computed as the average unsigned t value from our orientation-selective masks for the clockwise > counter-clockwise contrast applied to the orientation localizer data. The average t values for V1, V2 and V3 were numerically similar (see Author response image 1). However, the small variation in selectivity across areas was significant (Huynh-Feldt corrected F (32.76, 1.42) = 4.50, p =.029), which appears to be driven by slightly weaker selectivity in V3 compared to V2 or V1. We believe this small difference in orientation selectivity between areas is unlikely to have had a large impact on our results, especially given that we found no differences between visual areas in our main analysis.

**Author response image 1. respfig1:** Differences in orientation selectivity across areas. The average unsigned t value for the clockwise > counter-clockwise contrast that was applied to the orientation localizer data are plotted for all voxels within orientation-selective masks in V1, V2 and V3.

– Figure 3F: not all subjects had more activity if the stimulus was attended. Is that a reliable within subject effect, opposite for some subjects than what was expected? Or does this reflect noise in the quality of the data of individual subjects?

Figure 3F does not show difference in activity between attended and unattended conditions. Rather, it is an illustration of how consistently the result in Figure 3E was shown be individual subjects. Figure 3E plots average scores representing the extent to which the effects of attention and contrast were more strongly expressed in the agranular (deep and superficial) or granular (middle) layers. These were defined as the average effect of attention (or contrast) in the deep and superficial layers minus the effect in the middle layer. As such a positive score indicates a stronger agranular effect, while a negative score indicates a granular effect. Figure 3E shows that, on average, the effect of attention was significantly more agranular compared to stimulus contrast, which was more granular. Figure 3F then shows the difference between attention and contrast scores plotted in Figure 3E for each individual subject, where a positive score indicates that the attention effect was more agranular, and a negative score indicates that the contrast effect was more agranular for that subject. Overall, this plot is a demonstration of how consistent the grouper laminar profiles for attention and contrast reported in the rest of the figure were across individual subjects.

2) The authors frame their paradigm as a feature-based attention paradigm. However, the design of the stimulus is quite different to a typical feature-based attention paradigm and I doubt whether the participants required feature-based attention to solve the task. For example, if I'm being cued to attend to the clockwise bars, my strategy would be to fixate one of the bars of the appropriate color (e.g. white) and monitor that bar for width changes. Eye-movements weren't monitored as far as I can tell, but even if the participants did maintain fixation then this is still more reminiscent of a spatial-attention task. True feature-based attention requires the participant to attend to a feature in one part of space, and this modulates activity related to that feature in another part of space. The interpretation of the paradigm is not just of semantic interest but has a large-impact on much of the discussion and the relevance and impact of this work, so I would like to hear the authors thoughts on this.

The paradigm we used is the same paradigm described in the landmark paper by (Kamitani and Tong, 2005, cited >1,500 times). These authors refer to the paradigm as a feature-based attention paradigm, given that subjects were attending to one of two features of the plaid stimulus, each feature being a set of oriented bars. It does not appear possible to complete the task using only spatial attention because (1) the two features largely spatially overlap and (2) the phase of each element within the plaid is randomized between stimuli, meaning the spatial location of the bars themselves changes from trial to trial. It is true that we were not able to record eye movements during the fMRI experiment, but Kamitani and Tong ruled out the contribution of eye movements to the same paradigm using a control experiment. We have pasted their description of this control experiment below:

“We did an additional control experiment to address whether eye movements, orthogonal to the attended orientation, might account for the enhanced responses to the attended grating by inducing retinal motion. The visual display was split into left and right halves, and activity from corresponding regions of the contralateral visual cortex was used to decode the attended orientation in each visual field (Supplementary Figure 4). Even when the subject was instructed to pay attention to different orientations in the plaids of the left and right visual fields simultaneously, cortical activity led to accurate decoding of both attended orientations. Because eye movements would bias only one orientation in the whole visual field, these results indicate that the attentional bias effects in early visual areas are not due to retinal motion induced by eye movements.” (Kamitani and Tong, 2005, p. 683).

3) In the previous Current Biology paper the orientation-mask had quite an interesting spatial distribution in V1. Are these the same subjects? If not, was the same distribution observed?

Different subjects were used in the current paper. We observed a similar distribution to our previous study, with clockwise-preferring voxels being largely clustered in the inferior portion of V1, and counter-clockwise-preferring voxels mostly in the superior portion of V1. The distribution has also been observed and discussed in several other papers (e.g. Freeman et al., 2011; Swisher et al., 2010). We have added a new figure that shows the layout of orientation-selective voxels in V1 for a representative subject (Figure 3—figure supplement 8).

– In that paper in V1 bottom up signals are stronger in deep and superficial and weaker in L4 – their Figure 3B. Can you explain the difference?

The figure that the reviewer refers to (Figure 3B of Lawrence 2018) describes a rather different comparison, i.e. stimulus>baseline. In this case, there were in fact no differences between the different layers for this (*p = 0.648*), suggesting that the entire cortical column was equally activated by presenting a stimulus (vs. no stimulus). Here we show that increasing the contrast of the stimulus (high vs. low contrast comparison) leads to a stronger increase in the middle layer. In other words, here we describe a much subtler *modulation* of neural activity by increasing contrast, rather than a difference between a stimulus and a blank screen.

4) I think that the pooling across deep and superficial layers and then report "agranular layers" is misleading, given that the results for layer 4 are more similar to the deep than to the superficial layers. The dissimilarity between the low attention modulation in the deep layers in the present study with the previous monkey studies should be discussed.

The data we used to compute the agranular – granular scores plotted in Figure 3E are plotted in full in panels B and D, which show the strength of effects of attention and stimulus contrast within each layer separately. Exactly how the scores in panel E were computed is also clearly described in the figure legend, with reference to the data plotted in B and D. The pooling across agranular layers for the purposes of Figure 3E should therefore be clearly accessible to the reader, by inspecting the different panels of the same figure. Testing the extent to which bottom-up and top-down effects were expressed in the agranular versus the granular layers was directly relevant to our initial hypothesis. We discuss the discrepancies between our feature-based attention effect (which was strong in all layers, peaking in the superficial layers) and those reported in previous monkey studies on (spatial) attention. As the reviewer suggests, we now also highlight in this Discussion the differences in results with regards to the deep layer, specifically:

“In particular, Van Kerkoerle et al. report strong attentional modulations in the deep layers compared to the middle layer, which was not the case in our data.”

– The conclusion in the Discussion first paragraph "Moreover, our results pointed to stronger attention modulations in agranular cortical layers compared to contrast effects, which were strongest in the granular layer." seems to be a bit too optimistic to me.

Although the effect of attention did not significantly vary across cortical layers when considered in isolation, it was stronger in the agranular layers compared to a bottom-up effect of stimulus contrast, as we show in Figure 3E. It is this comparison that we intended to speak to in the quoted statement, and we now make this clearer in the manuscript:

“Moreover, by comparing the strength of attention and contrast modulation in agranular versus the granular layers, we found that attention effects were expressed more strongly in the agranular layers compared to effects from stimulus contrast, which were more granular.”

[Editors' note: further revisions were requested prior to acceptance, as described below.]

Reviewer #2:I am still a bit mixed about this paper. On the one hand, the authors convincingly addressed all issues that I had with the normalization.On the other hand, my issue with the grouping of superficial and deep layers into a "agranular compartment" was not addressed satisfactorily. As I mentioned in my first review, the attentional effects (or the ratio with bottom-up effects) in fact seem strongest in the superficial layers and weakest in the deep layers, and intermediate in layer 4. It does therefore not make much sense to average across the superficial and deep layers. The results are actually quite different from previous studies on spiking activity in monkeys, so that one wonders if laminar fMRI using the present methodology reflects the underlying neurophysiology.

We appreciate the reviewer’s point. We now mention more clearly, throughout the manuscript, that the attentional effect is most strongly present in the superficial layer (rather than the superficial and deep layers). The analysis of granular vs. agranular layers was a priori and theoretically motivated, in view of the fact that bottom-up input activates the granular layer 4 whereas top-down modulations avoid the granular layer but innervate the superficial and deep layers instead. We would therefore like to keep this analysis. We do however now spell out clearly throughout the manuscript that the difference in activity profile between the contrast and attention modulations is driven by a larger activity modulation in the superficial layers (attention) vs. middle layer (contrast).

The link with spiking activity in monkeys is difficult. To our knowledge, there hasn’t been a single study measuring layer-specific activity modulations due to feature-based attention in monkeys. Also, the link between spiking activity (reflecting output) and BOLD signal modulations (predominantly reflecting input) is not fully understood. For all these reasons, one cannot expect a 1:1 correspondence in findings between species and methods.

Several misleading statements remained in the revision:– In the Abstract: "top-down modulation is significantly stronger in deep and superficial layers than top down effects", which is not shown for the deep layers but only by using the misleading grouping into "agranular layers". I suspect that the effects are driven by the superficial layers.– Same at the end of the Introduction, final sentence.– In reality there are no clear differences in top-down effects between the layers (subsection “Dissociable laminar profiles of bottom-up and top-down response modulations”) but it is only if a comparison (subtraction) is made to the bottom-up effects.– This is also visible in Figure 4A, where the weakest attention effects are present in the deep layers, the strongest in the superficial layers and the granular layers are intermediate.– Discussion section: "We have shown that, in a task where bottom-up and top-down influences are manipulated independently, the overall BOLD response can be separated into top-down and bottom-up components by examining how these effects are organized across depth." I think that this is an overstatement. The only reliable laminar difference seems to be in the bottom up response across layers.– It should also be clarified consistently that these effects are driven by a difference in the contrast sensitivity rather than be a difference in the attention effects between layers.

We have adapted the statements to ensure that they do not mislead the readers and properly reflect the data. Specifically:

Abstract: “Bottom-up modulations were strongest in the middle cortical layer and weaker in deep and superficial layers, while top-down modulations were strongest in the superficial layers.”

Introduction: “As predicted, attentional modulations were more strongly expressed in agranular layers, particularly the superficial layers, while stimulus contrast modulations were largest in the granular layer”.

Results: “Therefore, it appears that top-down contributions to response modulations were stronger in the agranular layers compared to bottom-up contributions, which were strongest in the granular layer. As can be seen from Figure 3B, the agranular profile of attention was driven by the fact that the attentional modulation was strongest in the superficial layers.”

Discussion: “Moreover, by comparing the strength of attention and contrast modulation in agranular versus the granular layers, we found that attention effects were expressed more strongly in the agranular layers (specifically the superficial layers) compared to effects from stimulus contrast, which were more granular.”

Conclusion: “Top-down modulations from attention were overall stronger in agranular layers (specifically the superficial layers) compared to those from stimulus contrast, which were strongest in the granular layer.”

What is the rationale of the grouping of superficial and deep layers? Is it the wish to replicate the non-human primate studies?

We have explicated this in reply to point #1.

I would recommend that the three laminar compartments stay separate throughout the analysis (e.g. Figure 3E, F), and also in the Abstract and in the Discussion. It seems conceivable that in such an analysis with three laminar compartments, there is a difference in the ratio between top-down and bottom-up effects between superficial layers and the granular layers, but no such difference between the granular and deep layers. Such a discrepancy with the non-human primate work would also be a valuable outcome, and useful for future studies that plan to use laminar fMRI.

We thank the reviewer for this suggestion. The analysis in which the three laminar compartments are kept separate was already included in the manuscript [i.e., the modulation (attention/contrast) x depth (deep/middle/superficial) repeated measures ANOVA]. The fact that this analysis showed a significant modulation X depth interaction provides formal statistical support for the notion that there is a difference in the ratio between top-down and bottom-up effects between the layers. We now further unpacked this interaction by showing that the bottom-up effect varied significantly across depth (F [46, 2] = 8.43, p =.001), being largest at middle compared to deep (t [23] = 3.79, p =.001) and superficial (t [23] = 3.56, p =.002) depths. For the top-down effect, there was a statistical trend of activity differences between the layers (F [46, 2] = 2.82, p =.070). Unpacking this, we observed that the attention effect was significantly stronger in the superficial layers compared to the middle (t [23] = 2.11, p =.046) and deep layers (t [23] = 2.15, p =.042), while there was no difference in the strength of the attention effect between the deep and middle layers (t [23] = 0.36, p =.723). These post-hoc tests further corroborate the notion that is put forward by the reviewer, that the attentional effect is strongest in the superficial layers (rather than the deep layers). We have added these further post-hoc tests to the manuscript and we have tried to better qualify the nature of the attentional modulation throughout the manuscript.

Reviewer #3:Overall, I appreciate that the authors put in a tremendous amount of work to address all the questions from all three reviewers. I am satisfied with all the answers except this answer:"Our revised manuscript includes three additional figures displaying raw data to allow the reader to better assess the data quality. (1) A figure showing raw, layer-specific BOLD time courses for each experimental condition and each ROI (Figure 3—figure supplement 3)"This is a time course but not a map of actual layer activity.

We have now added a map of actual layer activity, Figure 3—figure supplement 9.